# Neutralizing antibody responses over time in a demographically and clinically diverse cohort of individuals recovered from SARS-CoV-2 acquisition in Africa: A cohort study

Nonhlanhla N. Mkhize[1,2], Shuying Sue Li[3], Jiani Hu[3], Samuel T. Robinson[3], Zaheer Hoosain[4], Nigel Garrett[5,6], Zvavahera M. Chirenje[7,8], Llewellyn Fleurs[9], Haajira Kaldine[1,2], Tandile Modise[1,2], Penny L. Moore[1,2,5], April K. Randhawa[3], Anton M. Sholukh[3], Julia Hutter[10], Laura Polakowski[11], Lawrence Corey[3], Katherine Gill[9], David C. Montefiori[11], Holly Janes[3], Shelly Karuna[3]\*, for the HVTN 405/HPTN 1901 Study Team[¶]

1 SAMRC Antibody Immunity Research Unit, University of the Witwatersrand, Johannesburg, South Africa, 2 National Institute for Communicable Diseases, National Health Laboratory Service, Johannesburg, South Africa, 3 Vaccine and Infectious Disease Division, Fred Hutchinson Cancer Research Center, Seattle, Washington, United States of America, 4 Josha Research Centre, Bloemfontein, South Africa, 5 Centre for the AIDS Programme of Research in South Africa (CAPRISA), University of KwaZulu-Natal, Durban, South Africa, 6 Department of Public Health Medicine, School of Nursing and Public Health, University of KwaZulu-Natal, Durban, South Africa, 7 University of Zimbabwe Clinical Trials Research Centre, Harare, Zimbabwe, 8 Department of Obstetrics, Gynecology and Reproductive Science, University of California San Francisco, San Francisco, California, United States of America, 9 Desmond Tutu HIV Centre, University of Cape Town, Cape Town, South Africa, 10 Division of AIDS, National Institute of Allergy and Infectious Diseases, National Institutes of Health, Bethesda, Maryland, United States of America, 11 Department of Surgery, Duke University Medical Center, Durham, North Carolina, United States of America

¶ Membership for the HVTN 405/HPTN 1901 Study Team is provided in the Acknowledgments.
\* mkaruna@fredhutch.org

## Abstract

COVID-19 has affected millions worldwide. Research characterized immune responses of individuals who acquired SARS-CoV-2 and identified co-factors, such as HIV, associated with greater likelihood of poor clinical outcomes. SARS-CoV-2-specific neutralizing antibodies (nAbs) are a strong correlate of protection but their elicitation in people living with HIV (PLWH), and particularly in southern Africa, is less well characterized. HVTN 405/HPTN 1901 was an observational cohort study of individuals recently recovered from SARS-CoV-2. We describe 323 participants enrolled early in the pandemic (June 2020 to January 2021) in Zambia (n = 12), Malawi (n = 13), Zimbabwe (n = 59), and South Africa (n = 239), profiling their SARS-CoV-2-specific nAb responses and associations with demographics, comorbidities, disease severity, and time since diagnosis based on linear and logistic regression. Participants' median age was 39 years, 63.5% were assigned female sex at birth, 71.2% were black African, and 39 (12.1%) were PLWH. Approximately one in four

**Data availability statement:** All relevant data are within the paper and its Supporting information files.

**Funding:** The HIV Vaccine Trials Network/ COVID-19 Prevention Network (HVTN/ CoVPN) is funded by the NIH/NIAID: UM1 AI068618 (Lab), UM1 AI068635 (Statistical and Data Management Center), UM1 AI068614 (Leadership and Operations Center). PLM is supported by the South African Research Chairs Initiative of the Department of Science and Innovation and the National Research Foundation (Grant No 98341). NNM, TM and PLM are supported by the SA Medical Research Council GIPD Program. The funders had no role in study design, data collection and analysis, decision to publish, or preparation of the manuscript.

**Competing interests:** The authors have declared that no competing interests exist.

participants (25.7%) had asymptomatic SARS-CoV-2, 47.4% were symptomatic but not hospitalized, and 26.9% were hospitalized with COVID-19. Participants in these groups were enrolled at a median of 51.5 days, 53 days, and 60 days post-SARS-CoV-2 diagnosis, respectively. SARS-CoV-2 nAbs were measured in serum using one of two calibrated assays. Most (291/322, 90.4%) participants had positive nAb responses at enrollment. Across all participants, nAb responses generally declined in magnitude between enrollment and 2–3 months thereafter, then increased through month 12 coincident with epidemiologically observed new waves of acquisition. In a multivariate model adjusted for potentially confounding factors, PLWH had a 65% lower geometric mean (GM) nAb ID50 titer compared to people without HIV (PWOH) (GMR: 0.35, p = 0.003, q = 0.006). Greater disease severity, older age (>55 years), high BMI (≥30) and diabetes were associated with higher nAb ID50 titers (all p < 0.05, all q < 0.20).These findings are important, as nAb titers are predictive of vulnerability to COVID-19.

## Introduction

With over 750 million reported cases of Coronavirus Disease 2019 (COVID-19) worldwide as of November 2024 [1], there remain opportunities to analyze and archive important data on the immune response to the causative agent, Severe Acute Respiratory Syndrome Coronavirus 2 (SARS-CoV-2). Acquisition of SARS-CoV-2 has been shown to manifest across a broad range of clinical severities, ranging from asymptomatic cases to severe disease, which can result in death. Early in the pandemic, characterizing the immune responses of convalescent individuals was a critical step in establishing benchmarks for vaccine-induced immunogenicity, and this remains relevant as vaccines are developed to target emerging viral strains.

SARS-CoV-2-specific neutralizing antibodies (nAbs) have been established as a correlate of vaccine-induced protection against COVID-19 [2,3]. Some natural history studies have also found binding or neutralizing antibodies induced by SARS-CoV-2 acquisition (note that acquisition is being used rather than infection in alignment with National Institute of Allergy and Infectious Diseases guidelines on non-stigmatizing language) to inversely correlate with subsequent reacquisition, although the strength of the findings has varied across studies [3–10]. Numerous studies have identified that prior infection plus vaccination, or "hybrid immunity", affords the strongest antibody response and the most effective protection [11–14]. Yet antibody responses to both vaccination and to SARS-CoV-2 acquisition are variable; thus, significant efforts have been undertaken to understand predictors of the nAb response and, by extension, immune protection. Comorbidities as well as clinical and demographic factors associated with different severities of COVID-19 are of interest, and several associations have been identified [15,16]. Notably, people living with HIV (PLWH) have been identified as a group with a higher likelihood of more severe outcomes [17–21], and yet comparative studies of their immune responses to acquisition are still limited and varied in their conclusions [22–26].

Approximately half of the estimated 39.9 million PLWH in 2023 resided in eastern and southern Africa [27]. For this and other reasons, the course of the COVID-19 pandemic in Africa has been unique [28–30]. Early reports suggested lower-than-expected rates of incidence and death [31–33], though this was likely influenced by inconsistencies in health infrastructure, including the availability of testing, reporting systems, and other resources, particularly early in the pandemic. This only exacerbated a persistent and pervasive lag of high-quality clinical data emanating from African populations.

Early in the pandemic, the hereon reported HVTN 405/HPTN 1901 trial was established with the goal of characterizing SARS-CoV-2-specific immunity in a multinational cohort of convalescent individuals. Results based on the subset of participants from the Americas (United States and Peru) have been described previously and demonstrated that nAb responses tended to peak 1 month after diagnosis and waned substantially over the following four months, albeit with considerable heterogeneity associated with demographic factors, pre-existing medical co-morbidities, and COVID-19 severity [34]. Peak antibody responses among PLWH with COVID-19 (i.e., symptomatic SARS-CoV-2 acquisition) were diminished compared to antibody responses among their counterparts without HIV (PWOH), and antibody responses among PLWH did not correlate with COVID-19 severity, as they did among PWOH [25].

In this report we considered participants enrolled in HVTN 405/HPTN 1901 in Africa: we characterized their nAb responses to SARS-CoV-2 acquisition and examined how these responses varied with COVID-19 severity, as well as demographic and clinical characteristics, including HIV status.

## Materials and methods

### Trial design

HVTN 405/HPTN 1901 was an observational cohort study (ClinicalTrials.gov NCT04403880) that enrolled 759 participants with recent SARS-CoV-2 acquisition at 53 clinical research sites in Malawi, South Africa, Zambia, Zimbabwe, Peru, and the United States [35]. This manuscript includes neutralizing antibody data for participants that enrolled in the southern Africa sites. Enrollment at these sites occurred between July 29, 2020, and March 31, 2021, and this analysis was further restricted to the 323 participants who were diagnosed prior to October 1, 2020. Written informed consent was obtained from each participant at enrollment using a standardized protocol consent form. Results pertaining to participants from Peru and United States have been reported elsewhere [25,34]. Participants were stratified by COVID-19 symptom severity (asymptomatic, symptomatic outpatient, or hospitalized) and by age (18–55 or >55 years of age). For symptomatic outpatient and hospitalized participants, SARS-CoV-2 diagnosis dates were based on the date of the first positive SARS-CoV-2 nucleic acid amplification test (NAAT) or the date of symptom onset, whichever occurred earlier. Symptomatic outpatient participants and participants hospitalized due to COVID-19 were eligible to enroll 1–8 weeks after disease resolution. Participants who were asymptomatic, despite confirmed SARS-CoV-2 acquisition, defined by a reported positive SARS-CoV-2 test (PCR or antigen), were eligible to enroll 2–10 weeks after diagnosis. Additional information on study inclusion and exclusion criteria is included in S1 Text.

Demographic and clinical data were collected for all participants at a required baseline study visit, and at optional follow-up visits 2, 4, and 12 months after enrollment. Medical history (including HIV and COVID-19 history) was documented by the enrolling clinic using participant health records. Blood and nasopharyngeal samples were collected for virologic and immunologic assays. Ethics Committee or Institutional Review Board (IRB) approval was granted by a Central IRB in the United States (Advarra IRB) and, as applicable, by individual review boards and applicable regulatory agencies for the clinical research sites.

### COVID severity grading

COVID-19 severity was defined using a custom grading scale that followed the Division of AIDS (DAIDS), Food and Drug Administration (FDA), and World Health Organization (WHO) grading scales as closely as possible given the constraints posed by the COVID-19 symptom and treatment data that were collected. From least to most severe, severity at the time

of enrollment was based on the following scale: 1: Asymptomatic, 2: Symptomatic, not hospitalized, 3: Hospitalized, no supplemental oxygen, 4: Hospitalized, with supplemental oxygen, but no ICU or intubation, 5: Hospitalized, with ICU or intubation, 6: Hospitalized, with ECMO (extracorporeal membrane oxygenation). Severity categories were mutually exclusive and were assessed from highest severity to lowest. For the purposes of analysis, participants with grades 3–6 were combined to form a single "hospitalized" severity group.

### Neutralizing antibody detection

For PWOH, nAbs against SARS-CoV-2 were measured in 293T/ACE2 cells as a function of reduction in Tat-induced luciferase (Luc) reporter gene expression after a single round of infection with lentivirus particles pseudotyped with the SARS-CoV-2 Wuhan-1/D614G Spike protein [36]. SARS-CoV-2 pseudotyped lentiviruses were prepared by co-transfecting 293T cells with the SARS-CoV-2 Spike plasmid, a firefly luciferase-reporter gene plasmid, a TMPRSS-2-expressing plasmid, and a lentivirus backbone plasmid. Pseudovirions were titrated for infectivity and assayed for neutralization in 293T/ACE2 cells. Luciferase activity was quantified by luminescence and was directly proportional to the number of infectious virus particles present in the test samples.

In PLWH on antiretroviral therapy (ART), the ART interferes with the 293T/ACE2 lentivirus assay. Thus, for PLWH, nAbs were measured by the vesicular stomatitis virus (VSV) pseudovirus neutralization assay described by Sholukh et al (S2 Text) [37]. VSV pseudovirus was prepared using a codon optimized gene of SARS-CoV-2 Spike protein (YP_009724390.1) cloned into a pcDNA3.1 (PsVSVLucD19) and VSV(*G*∆G-luciferase) system purchased from Kerafast (Boston, MA) [38,39].

Neutralization titers were defined as the inhibitory dilution of serum samples at which relative light units (RLUs) were reduced by either 50% (ID50) or 80% (ID80) compared to virus control wells. A positive response was defined as neutralization ≥50% at the lowest serum dilution tested (1:10). For negative responses, the ID50 (ID80) titer was set to 5.

### Statistical analysis

Neutralizing antibody responses for the 293T/ACE2 lentivirus and VSV assays were calibrated prior to analysis to facilitate comparisons across PLWH/PWOH groups (S2 and S3 Text, and S1 Fig).

Regression models were used to associate nAb responses at enrollment with individual participant baseline characteristics: COVID-19 severity, age (>55 vs 18–55), body mass index (BMI) (≥30 vs <30), HIV status, prolonged viral shedding (detection of SARS-CoV-2 on two tests at least 21 days apart), preexisting medical conditions (hypertension; asthma, COPD, or emphysema; and diabetes), smoking history, sex assigned at birth, and days since SARS-CoV-2 diagnosis. Models were adjusted for a pre-specified set of covariates that have been previously found to associate with nAb titers COVID-19 severity, age, sex at birth, African region (RSA vs. non-RSA), and days since SARS-CoV-2 diagnosis. We refer to these as "adjustment factors". nAb response titers were modeled on the log scale using linear regression, and nAb response positivity was modeled using Firth logistic regression [40].

The Benjamini and Hochberg method [41] of controlling the false discovery rate was used to adjust for multiple comparisons. A false-discovery rate, or q-value, of less than 0.20 was considered statistically significant.

To describe the trajectory of nAb responses over time since SARS-CoV-2 diagnosis by COVID-19 severity and HIV status, a generalized additive mixed model (GAMM) was used [42]; nAb responses were modeled on the log scale. The R mgcv package was used with order 2 of penalty derivative and basis dimension 50.

All analyses were done using R [43].

## Results

### Study population

A total of 323 participants from four countries were enrolled in the African cohort between 29 July 2020 and 18 January 2021, diagnosed prior to 1 October 1, 2020, and included in this analysis (Table 1). The majority of participants were from

**Table 1. Participant characteristics at the time of enrollment for the Africa cohort.**

| | Total (N = 323) | Asymptomatic (N = 83) | Symptomatic, Not Hospitalized (N = 153) | Hospitalized (N = 87) |
|---|---|---|---|---|
| **Age** | | | | |
| Mean (SD) | 40.9 (13.43) | 36.8 (14.43) | 41.3 (12.67) | 44 (12.92) |
| Median (IQR) | 39 (31, 51) | 32 (26, 46.5) | 40 (32, 50) | 43 (34, 55) |
| Range | 18 - 84 | 18-76 | 19-84 | 20-72 |
| **Age Range, n (%)** | | | | |
| 18 - 55 | 265 (82.0%) | 69 (83.1%) | 130 (85%) | 66 (75.9%) |
| >55 | 58 (18.0%) | 14 (16.9%) | 23 (15%) | 21 (24.1%) |
| **Sex Assigned at Birth, n (%)** | | | | |
| Female | 205 (63.5%) | 40 (48.2%) | 107 (69.9%) | 58 (66.7%) |
| Male | 118 (36.5%) | 43 (51.8%) | 46 (30.1%) | 29 (33.3%) |
| **HIV Status, n (%)** | | | | |
| People Living with HIV (PLWH) | 39 (12.1%) | 7 (8.4%) | 13 (8.5%) | 19 (21.8%) |
| People Without HIV (PWOH) | 284 (87.9%) | 76 (91.6%) | 140 (91.5%) | 68 (78.2%) |
| **Race** | | | | |
| White | 9 (2.8%) | 1 (1.2%) | 7 (4.6%) | 1 (1.1%) |
| Black | 230 (71.2%) | 65 (78.3%) | 103 (67.3%) | 63 (72.4%) |
| Asian | 10 (3.1%) | 1(1.2%) | 6 (3.9%) | 3 (3.4%) |
| Other* | 72 (22.3%) | 16 (19.3%) | 36 (23.5%) | 20 (23%) |
| **Ethnicity** | | | | |
| Hispanic or Latinx | 1 (0.3%) | 0 (0.0%) | 0 (0.0%) | 1 (1.1%) |
| Not Hispanic or Latinx | 322 (99.7%) | 83 (100%) | 153 (100%) | 86 (98.8%) |
| **BMI** | | | | |
| Mean (SD) | 30.8 (8.36) | 28.7 (8.31) | 30.8 (7.84) | 32.7 (8.9) |
| Median (IQR) | 29.2 (24.9, 34.9) | 26.9 (22.1, 33) | 29.4 (25.8, 34.8) | 30.4 (26.8, 37.1) |
| Range | 15.2 - 65.1 | 17 - 55.5 | 15.2 - 57.9 | 16.7 - 65.1 |
| **BMI Category, n (%)** | | | | |
| <30 | 171 (52.9%) | 53 (63.9%) | 78 (51%) | 40 (46%) |
| ≥30 | 148 (45.8%) | 30 (36.1%) | 72 (47.1%) | 46 (52.9%) |
| **Days Since SARS-CoV-2 Diagnosis** | | | | |
| Mean (SD) | 54.1 (20.19) | 50.6 (15.36) | 51.4 (20.84) | 62.1 (21.07) |
| Median (IQR) | 55 (38, 66) | 51.5 (38, 62) | 53 (34.5, 65) | 60 (46.5, 76) |
| Range | 14 - 185 | 14 - 90 | 15 - 185 | 22 - 137 |
| **Days Since SARS-CoV-2 Diagnosis Range Category** | | | | |
| <28 | 24 (7.4%) | 5 (6%) | 17 (11.1%) | 2 (2.3%) |
| 28 -<42 | 68 (21.1%) | 19 (22.9%) | 37 (24.2%) | 12 (13.8%) |
| 42 -<56 | 70 (21.7%) | 23 (27.7%) | 24 (15.7%) | 23 (26.4%) |
| 56+ | 158 (48.9%) | 35 (42.2%) | 73 (47.7%) | 50 (57.5%) |
| **Medical History, n (%)** | | | | |
| Currently smoke cigarettes or marijuana | 31 (9.6%) | 10 (12%) | 16 (10.5%) | 5 (5.7%) |
| Ever smoked cigarettes or marijuana | 62 (19.2%) | 19 (22.9%) | 33 (21.6%) | 10 (11.5%) |
| Hypertension | 77 (23.8%) | 13 (15.7%) | 30 (19.6%) | 34 (39.1%) |
| Asthma, COPD, or Emphysema | 16 (5%) | 2 (2.4%) | 9 (5.9%) | 5 (5.7%) |
| Diabetes | 45 (13.9%) | 7 (8.4%) | 18 (11.8%) | 20 (23%) |
| Prolonged Viral Shedding | 5 (1.5%) | 1 (1.2%) | 2 (1.3%) | 2 (2.3%) |

*"Other" Race was an option on the case report form; participants were asked to specify, and responses included the following: African (n = 17), Black African or African Black or SA black (n = 28), Colored (n = 8), Indian or SA Indian (n = 15), Mixed race (n = 2), and Nsundu (n = 1).

South Africa (RSA) (n = 239) compared to 84 participants from the other African countries (Zambia (n = 12), Malawi (n = 13), Zimbabwe (n = 59)). Across all participants, more experienced symptomatic COVID-19 without hospitalization (n = 153, 47.4%), compared to those with asymptomatic acquisition (n = 83; 25.7%), or requiring hospitalization (n = 87; 26.9%). The participant pool was diverse: 63.5% were assigned female sex at birth, the median age was 39 years (interquartile range [IQR]: 31–51, range: 18–84), 18.0% were 55 years of age or older, 71.2% were black African, and multiple comorbidities were well represented. Across all severities, 39 (12.1%) participants were PLWH, of which 7 (17.9%) were asymptomatic, 13 (33.3%) were symptomatic but did not require hospitalization, and 19 (48.7%) were hospitalized.

Descriptively, COVID-19 severity was associated with several participant characteristics (Table 1). Participants 18–55 years old constituted a larger proportion of those in the symptomatic non-hospitalized category (85.0%) than in the asymptomatic (83.1%) and hospitalized (75.9%) groups, whilst adults >55 constituted a larger portion of the hospitalized (24.1%) group than the asymptomatic (16.9%) and symptomatic non-hospitalized (15.0%) groups. Participants assigned female sex at birth were represented more frequently in the symptomatic non-hospitalized (69.9%) and hospitalized (66.7%) groups compared to the asymptomatic group (48.2%). PLWH (21.8%) and individuals who had hypertension (39.1%) were more frequent in the hospitalized group compared to the other groups (asymptomatic, 8.4% PLWH and 15.7% with hypertension; symptomatic, 8.5% PLWH and 19.6% with hypertension).

Participants who were asymptomatic were enrolled a median of 51.5 days after SARS-CoV-2 acquisition diagnosis (IQR: 38–62 days), while symptomatic, non-hospitalized participants were enrolled a median of 53 days post-diagnosis (IQR: 34.5 to 65 days) and hospitalized participants were enrolled a median of 60 days post-diagnosis (IQR: 46.5 to 76 days).

### Descriptive analysis of neutralizing antibody responses at enrollment

Most participants had positive SARS-CoV-2 nAb responses at enrollment: response rates were 87.95% (95% confidence interval (CI): 79.22%, 93.32%) for asymptomatic participants, 90.13% (95% CI: 84.36%, 93.93%) for symptomatic non-hospitalized participants, and 93.10% (95% CI: 85.76%, 96.80%) for hospitalized participants (S1 Table, Fig 1). Response magnitudes were more variable, with geometric mean (GM) ID50 titers of 123.5 (95% CI: 81.0, 188.4), 296.9 (211.6, 416.7), and 547.5 (95% CI: 362.3, 827.5) in the asymptomatic, symptomatic, and hospitalized groups, respectively. Response rates were high (>80%) across all subgroups of participants defined by baseline participant characteristics (S1 Table, Fig 1). GM ID50 titers ranged from a low of 85.6 (95% CI: 4.4, 1666.8) in the five participants with prolonged viral shedding, (next lowest was 110.5 [95% CI: 56.0, 218.0] among 31 current smokers), to a high of 790.4 (95% CI: 493.9, 1265.0) among 45 participants with diabetes. Numerically, higher nAb titers were observed for individuals >55 years old vs. 18–55 (GM ID50: 719.0 [95%CI: 484.6, 1066.9] vs. 228.0 [95%CI: 175.9, 295.5]), without prolonged viral shedding (GM ID50: 284.7 [95%CI: 226.2, 358.3] vs. 85.6 [95%CI: 4.4, 1666.8]) and with diabetes (ID50: 790.4 [95%CI: 493.9, 1265.0] vs. 236.0 [95%CI: 183.7, 303.1])

### Association between participant characteristics and neutralizing antibody responses at enrollment

Regression analyses revealed numerous factors independently associated with anti-SARS-CoV-2 nAb response positivity and ID50 titer at enrollment (Table 2, Fig 1). After controlling for the adjustment factors, nAb response rates did not differ by COVID-19 severity (p = 0.546, q = 0.728), though nAb titers did: participants who were hospitalized with COVID-19 symptoms exhibited over 4-fold higher geometric mean nAb ID50 titers compared to asymptomatic participants (geometric mean ratio [GMR] = 4.62, 95% CI: 2.51, 8.51; p < 0.001, q < 0.001) and 1.73-fold higher geometric mean nAb titer compared to symptomatic participants who were not hospitalized (95% CI for GMR: 1.01, 2.96; p = 0.046, q = 0.066). Symptomatic participants who were not hospitalized had 2.67-fold higher geometric mean nAb titer compared to asymptomatic participants (95% CI for GMR: 1.56, 4.56; p < 0.001, q = 0.001).

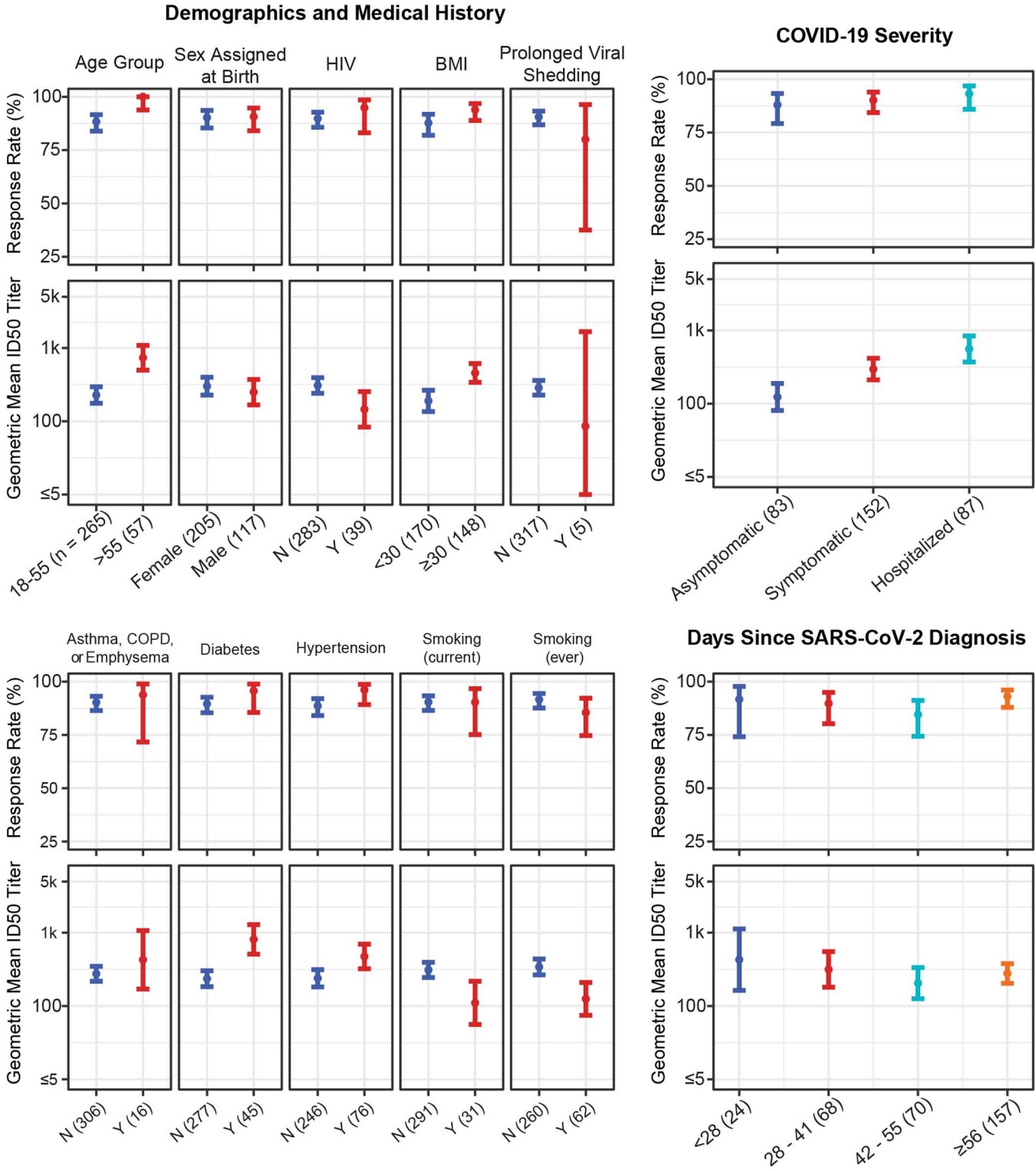

**Fig 1. Estimated SARS-CoV-2 neutralizing antibody (nAb) response rate and geometric mean (GM) nAb ID50 titers at enrollment by select baseline participant characteristics.** Points show estimated response rates and GM ID50 titers; lines show 95% confidence intervals. Y = yes; N = no. The number of participants in each category is given in parentheses in the labels along the x-axis.

**Table 2. Results of multivariate models associating baseline participant characteristics with SARS-CoV-2 neutralizing antibody (nAb) ID50 titer among all participants at enrollment.** Each predictor of interest was studied independently for its association with a positive nAb response (using Firth logistic regression) and with the nAb ID50 titer (using linear regression). Each regression model adjusts for the following adjustment factors selected based on prior literature: COVID-19 severity, age, sex at birth, African region, and days since SARS-CoV-2 acquisition diagnosis. (OR = Odds Ratio; GMR = Geometric Mean Ratio).

| | Positive nAb Response | | | | nAb ID50 titer | | | |
|---|---|---|---|---|---|---|---|---|
| | OR | 95% CI | P-Value | Q-Value | GMR | 95% CI | P-Value | Q-Value# |
| Age (>55 vs. 18–55) | 14.02* | [1.90, 1789.08] | **0.004** | **0.036** | 2.91 | [1.65, 5.14] | **<0.001** | **0.001** |
| BMI (≥30 vs <30) | 2.06 | [0.85, 5.23] | 0.111 | 0.410 | 2.12 | [1.32, 3.40] | **0.002** | **0.005** |
| HIV (Yes vs. No) | 1.97 | [0.56, 10.44] | 0.317 | 0.719 | 0.35 | [0.18, 0.69] | **0.003** | **0.006** |
| COVID-19 Severity | | | 0.546 | 0.728 | | | **<0.001** | **<0.001** |
| Symptomatic (not hospitalized) vs. Asymptomatic | 1.59 | [0.63, 3.92] | 0.316 | 0.719 | 2.67 | [1.56, 4.56] | **<0.001** | **0.001** |
| Hospitalized vs. Symptomatic (not hospitalized) | 0.99 | [0.36, 2.95] | 0.989 | 0.991 | 1.73 | [1.01, 2.96] | **0.046** | **0.066** |
| Hospitalized vs. Asymptomatic | 1.58 | [0.53, 4.98] | 0.412 | 0.719 | 4.62 | [2.51, 8.51] | **<0.001** | **<0.001** |
| Asthma, COPD, or Emphysema (Yes vs. No) | 0.93 | [0.20, 8.90] | 0.933 | 0.991 | 1.18 | [0.44, 3.19] | 0.738 | 0.814 |
| Diabetes (Yes vs. No) | 1.01 | [0.29, 5.28] | 0.991 | 0.991 | 2.17 | [1.14, 4.16] | **0.019** | **0.031** |
| Hypertension (Yes vs. No) | 1.57 | [0.52, 6.29] | 0.449 | 0.719 | 1.03 | [0.59, 1.83] | 0.908 | 0.908 |
| Cigarettes or Marijuana Smoker (Current) (Yes vs. No) | 0.63 | [0.19, 2.67] | 0.497 | 0.723 | 0.34 | [0.16, 0.72] | **0.005** | **0.010** |
| Cigarettes or Marijuana Smoker (Ever) (Yes vs. No) | 0.45 | [0.18, 1.21] | 0.111 | 0.410 | 0.34 | [0.19, 0.60] | **<0.001** | **0.001** |
| Days Since SARS-CoV-2 Diagnosis (Per day) | 1.02 | [1.00, 1.04] | 0.128 | 0.410 | 1.00 | [0.99, 1.01] | 0.673 | 0.814 |
| Sex Assigned at Birth (Male vs. Female) | 1.41 | [0.62, 3.42] | 0.421 | 0.719 | 1.07 | [0.67, 1.72] | 0.763 | 0.814 |
| Race (Non-Black vs. Black) | 0.87 | [0.36, 2.28] | 0.766 | 0.942 | 0.86 | [0.53, 1.41] | 0.560 | 0.747 |

# Q-value is the false discovery rate (FDR)-adjusted p-value for multiple comparisons.

*Large odds ratio and uncertainty thereof is due to the observation of a 100% response rate for those >55 years of age.

Too few participants exhibited prolonged viral shedding to make formal comparisons of this characteristic.

After controlling for the adjustment factors, participants who had ever been cigarette or marijuana smokers, or were currently cigarette or marijuana smokers, had lower anti-SARS-CoV-2 GM ID50 titer at enrollment (GMR=0.34, 95% CI: 0.16, 0.72; p=0.005, q=0.010 for current smokers; GMR=0.34, 95% CI: 0.19, 0.60, p<0.001, q=0.001 for ever smokers). Older participants (age>55 vs 1855) had significantly higher nAb response rates and GM ID50 titers (response odds ratio (OR) = 14.14, 95% CI: 1.90, 1789.08, p=0.004, q=0.036; GMR=2.91, 95% CI: 1.65, 5.14, p<0.001, q=0.001). Participants with high BMI (≥30 vs<30; p=0.002, q=0.005) and participants living with diabetes (Yes vs No; p=0.019, q=0.031) had more than double the GM ID50 titers of their lower BMI or diabetes-free counterparts (BMI: GMR=2.12, 95% CI: 1.32, 3.40; diabetes: GMR=2.17, 95% CI: 1.14, 4.16) (Table 2). Sex assigned at birth, race, and some medical comorbidities (hypertension; and any of COPD, emphysema, or asthma) were not associated with ID50 titers.

Results were generally similar and conclusions unchanged based on an analysis of ID80 nAb titers (S2 Table).

**Neutralizing antibody kinetics relative to COVID-19 waves**

Neutralizing antibody kinetics were evaluated with measurement of titers from samples collected at Visit 1 (enrollment) and Visits 2, 3, and 4, corresponding to approximately 2, 3, and 12 months after enrollment (Fig 2; S2 Table). Among both PLWH and PWOH and in all COVID-19 severity groups, nAb titers tended to decline from Visit 1 to Visit 2. There tended to be an increase in neutralization titers between Visits 2 and 3, and an even higher increase from Visit 3 to Visit 4.

Population-average trajectories in SARS-CoV-2 nAb responses were considered in the context of waves of SARS-CoV-2 acquisition in the communities where the study was conducted (Fig 3). nAb ID50 titers generally decreased between Visit 1 and the subsequent 100 days, after which there was a steady increase. The increase in nAb responses observed between Visits 3 and 4, at approximately 12 months after enrollment, coincided with the large Delta COVID-19

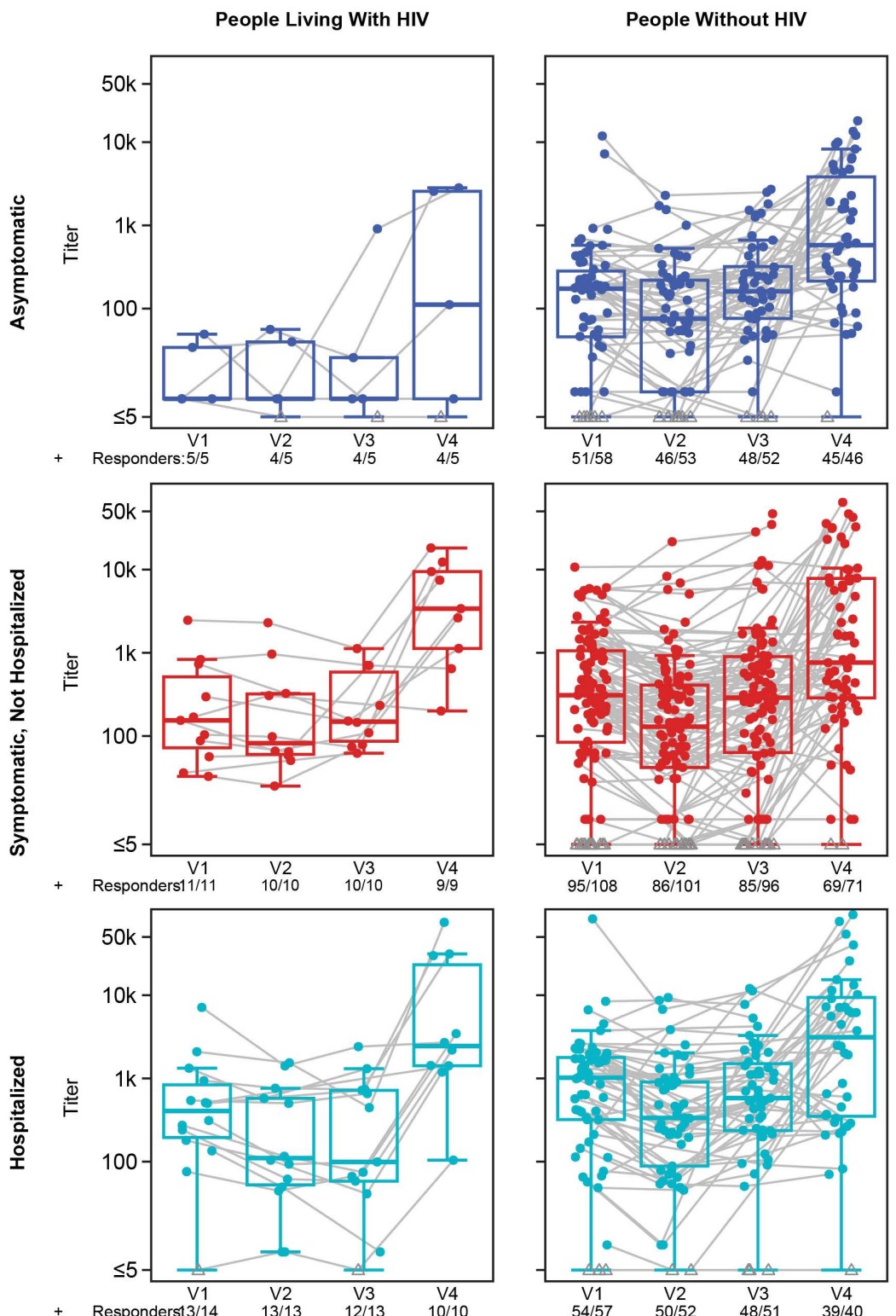

**People Living With HIV** — **People Without HIV**

**Fig 2. Distribution and traces of individual SARS-CoV-2 neutralizing antibody (nAb) responses over time by participant HIV status and COVID-19 severity. nAb ID50 titers were measured at enrollment (V1) and approximately 2 months (V2), 4 months (V3), and 12 months (V4) thereafter.** Boxplots indicate the median (middle bar), interquartile range (box length), the most extreme data points that are no more than 1.5 times the interquartile range, or if no value meets this criterion, to the data extremes (whiskers).

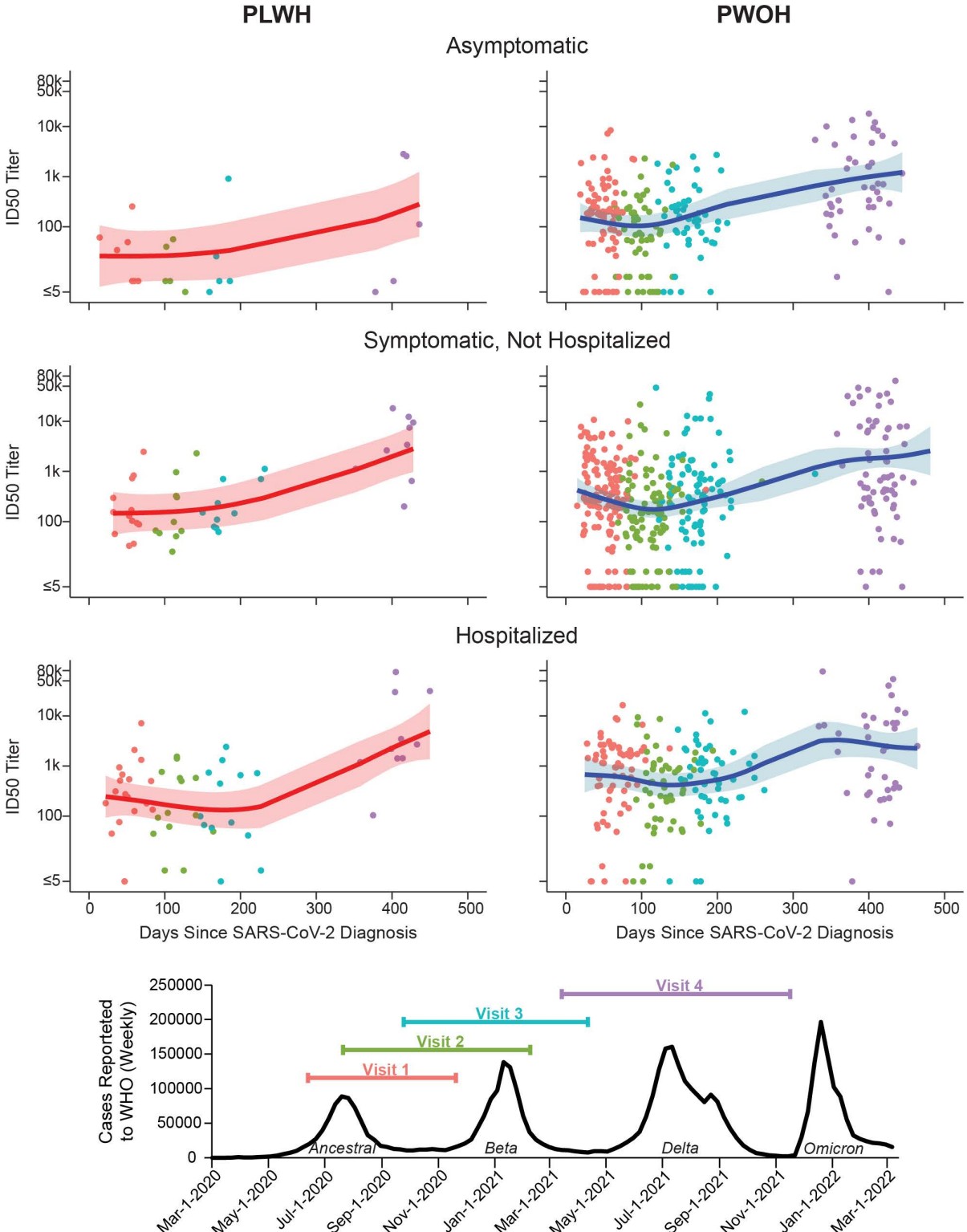

**Fig 3. Average SARS-CoV-2 neutralizing antibody trajectory by HIV status and COVID-19 severity group.** Individual nAb ID50 titers are shown as well as population geometric mean trajectories and 95% confidence bands estimated using a generalized additive mixed model (GAMM). Bottom

panel shows the total number of COVID-19 cases reported to the WHO in the countries included in this analysis over calendar time, with the study visit intervals overlaid. Symptomatic participants and participants hospitalized due to COVID-19 were eligible to enroll 1-8 weeks after COVID-19 disease resolution. Asymptomatic participants were eligible to enroll 2-10 weeks after SARS-CoV-2 acquisition diagnosis. Visit 1 was the enrollment visit. Visits 2, 3, and 4 occurred at 2, 4, and 12 months after enrollment, respectively.

wave in southern Africa in the second half of 2021 (Fig 3). In addition, starting in February 2021, southern African countries started their national COVID-19 vaccination programs, coinciding with Visit 3 and Visit 4 windows.

### Neutralizing antibodies in participants with and without HIV

While positive nAb response rates were similar between PLWH and PWOH (Odds Ratio [OR] = 1.97 [95% CI: 0.56, 10.44], p = 0.317, q = 0.718; Table 2), the 39 PLWH had an estimated 65% lower geometric mean anti-SARS-CoV-2 nAb ID50 titer at enrollment compared to PWOH (GMR = 0.35 [95% CI: 0.18, 0.69], p = 0.003, q = 0.006), after adjusting for age, sex assigned at birth, region, days since SARS-CoV-2 diagnosis, and COVID-19 severity. This difference in the GM ID50 titer by HIV status was apparent among asymptomatic (PLWH = 28.3 [95%CI: 8.4, 95.1] vs. PWOH = 141.1 [95%CI: 91.0, 219.9]), symptomatic (PLWH = 168.3 [95%CI: 77.8, 363.9] vs. PWOH = 313.1 [95%CI:217.5, 450.7]), and hospitalized (PLWH = 240.3 [95%CI: 100.5, 574.9] vs. PWOH = 689.1 [95%CI: 432.8, 1097.3]) participants (S3 Table).

These trends between PLWH and PWOH generally persisted over time: PLWH and PWOH continued to exhibit similar response rates, and PLWH had lower or comparable titers across Visits 2, 3, and 4. At Visit 2, approximately 2 months after enrollment, the GM ID50 titer was 15.0 (95% CI: 4.0, 57.0) among asymptomatic PLWH vs. 70.4 (95% CI: 44.2, 112.2) among PWOH, 153.0 (95% CI: 54.4, 430.2) vs. 129.6 (95% CI: 82.9, 202.8) among symptomatic PLWH vs. PWOH, and 150.1 (95% CI: 55.3, 407.6) vs. 322.0 (95% CI: 200.1, 518.2) among hospitalized PLWH vs. PWOH (S3 Table). At Visit 3, approximately 3–4 months after enrollment, GM ID50 titers among PLWH vs. PWOH were 24.0 (95% CI: 1.7, 333.1) vs. 151.8 (95% CI: 98.7, 233.7) among asymptomatic; 202.3 (95% CI: 95.4, 428.9) vs. 252.4 (95% CI: 156.6, 406.7) among symptomatic, non-hospitalized; and 152.5 (95% CI: 47.6, 489.3) vs. 529.7 (95% CI: 325.4, 862.3) among hospitalized individuals. Thus, while both groups experienced elevations in their ancestral strain antibodies during follow-up, titers in PLWH generally stayed lower.

Results were similar and conclusions unchanged based on an analysis of ID80 titers (S4 Table).

### Discussion

This work characterized the SARS-CoV-2-specific nAb response of a diverse population across four countries in southern Africa (South Africa, Zimbabwe, Zambia, and Malawi) shortly after SARS-CoV-2 diagnosis during the early stages of the pandemic (between June 2020 and January 2021). Importantly, this study evaluated nAb responses elicited after primary SARS-CoV-2 acquisition, stratified according to COVID-19 severity and emphasizing comparisons between participants living with and without HIV.

NAb response rates at enrollment were consistently high across baseline characteristics, ranging from 80% to a high of 100% among the 58 individuals over age 55. Consistent with convalescence from acquisition approximately 6 weeks prior, geometric mean nAb titers at enrollment were more variable between groups, with age > 55 years old, disease severity, diabetes, high BMI, and not smoking associated with higher magnitude responses. Consistent with the results from the Americas cohort [34], most participants exhibited a steady decrease in nAb titers between enrollment and the next visit, approximately 100 days thereafter, although this trend was less pronounced in the cohort of PLWH. Beyond that and through 12 months (Visit 4), nAb titers generally increased over time, which suggests new antigenic stimulation given that antibodies would be expected to exhibit continual decline [34,44,45]. This phenomenon could be the result of a number of

external factors, most notably the continuing waves of acquisition and the beginning of vaccine rollout. There were appreciable peaks in SARS-CoV-2 acquisition rates based on population-level epidemiological data [1] for the contributing countries; these peaks corresponded to the Beta SARS-CoV-2 variant during the timeframe of Visits 2 and 3 (around 2 and 4 months, respectively), and to the Delta variant during the timeframe of Visit 4 (around 12 months). Clinical resources (e.g., test kits) were still strained during these times, so confirmatory testing was rare.

Many of our findings are consistent with those published from the Americas cohort of this same trial [34], and with the broader literature, though some differences were observed. In particular, the associations we found between nAb responses to SARS-CoV-2 acquisition and disease severity, diabetes, BMI, and age have also been documented elsewhere [15,34,46]. In contrast, the lack of an association with sex assigned at birth and hypertension contrasts with several published reports [34,47–49]. The consistency between our results and prior literature is of particular importance in the context of an overall dearth of SARS-CoV-2 immunogenicity data published from trials conducted in southern Africa. It is noteworthy that the factors above remain associated with nAb responses irrespective of geography and a myriad of other societal, cultural, and potentially clinical factors.

The cohort of PLWH acutely infected with SARS-CoV-2 in southern Africa was a unique strength of this study. While SARS-CoV-2-specific nAb response rates were similar between PLWH and PWOH, PLWH had nAb ID50 titers that were, on average, approximately one-third lower than those of PWOH at enrollment, consistent with data from the Americas cohort of this study showing reduced immunogenicity in PLWH [25]. Interestingly, this effect was not solely driven by any single severity category, as both extremes (asymptomatic and hospitalized) showed significant differences between PLWH and PWOH. Across follow-up visits there was more variability, and it was not possible to disentangle potential differences between PLWH and PWOH resulting from waning immunity, possible reacquisition, or early vaccination. The variability at later time points may have been influenced by potential differences in COVID-19 vulnerability and vaccine uptake in PLWH. Nevertheless, PLWH who were asymptomatic tended to have lower nAb titers than asymptomatic PWOH across all four study visits.

The results from this study must be viewed in the context of various limitations. Some of these limitations were a consequence of the timing of the trial: the study protocol was written in the earliest stages of the pandemic and conducted when there was still pervasive uncertainty surrounding SARS-CoV-2 acquisition and pathogenesis. Healthcare resources and infrastructure were stressed, resulting in considerable constraints and inconsistencies in practices at clinical facilities across the globe. Specific to this cohort, the threshold for hospitalization was likely inconsistent, as hospitals needed to prioritize the most severe cases and admittance would therefore have been subject to local case fluctuations and capacity. Nonessential person-to-person contact was minimized as a part of pandemic-related protections instituted by the study or clinical research institutional policies, and as a result some study procedures were conducted by phone and in-person portions were often substantially time-limited, which limited more comprehensive data collection. Data regarding post-enrollment SARS-CoV-2 reacquisition was rarely collected due to limited accessibility of SARS-CoV-2 testing kits and policies limiting the type of allowable research clinic procedures early in the pandemic, and vaccine access and reporting was inconsistent during the study period. These limitations impaired our ability to precisely characterize the impacts of reacquisition and vaccination on neutralizing antibody trajectories, particularly through the last two study visits that took place during or after the Beta and Delta waves and the gradual introduction of vaccines. Similarly, relevant HIV data, including viral loads, CD4 counts, and antiretroviral status were rarely available due to pandemic-related stress on healthcare infrastructure and concomitant disruptions in the conduct of otherwise routine follow-up care, including viral load monitoring and sharing of medical records. These HIV-related data would be useful in more accurately characterizing the PLWH cohort. Though we do identify distinct elements of the immunologic profile in PLWH compared to PWOH, this remains an incompletely resolved area of interest, as some studies have reported similar SARS-CoV-2-specific antibody responses in PLWH and PWOH [23,25,50–52], while others have reported reduced responses in PLWH [25,53]. Furthermore, as an observational cohort study, the differences we observed in nAb responses between groups may be subject to

bias due to confounding; while analyses adjusted for factors measured and anticipated to affect nAb responses, there may be others that were not controlled for or measured. There was also a survivorship bias in the study population, as participants were only eligible to enroll after disease resolution or at least two weeks after confirmed asymptomatic SARS-CoV-2 acquisition.

Altogether, these results provide an important characterization of neutralizing antibody responses to primary SARS-CoV-2 acquisition—a unique endpoint as SARS-CoV-2-naïve individuals are an increasing rarity, globally—immediately following recovery and over time in southern Africa early during the COVID-19 pandemic. Factors explaining variability in the antibody responses align well with those identified in other populations and regions of the world. Moreover, the results provide additional evidence that people living with HIV in Africa do mount a more limited neutralizing antibody response to SARS-CoV-2 acquisition, suggesting timely boosters and additional precautions to limit exposure may be prudent.

## Supporting information

**S1 Text. Study inclusion and exclusion criteria.**
(DOCX)

**S2 Text. Vesicular stomatitis virus (VSV) antibody assay details.**
(DOCX)

**S3 Text. Calibration of neutralizing antibody assays.**
(DOCX)

**S1 Fig. Scatterplot of neutralizing antibody (nAb) responses from the VSV assay before and after calibration vs. neutralizing antibody (nAb) responses for the 293T/ACE2 assay, based on the calibration cohort.**
(DOCX)

**S1 Table. Estimated anti-SARS-CoV-2 neutralizing antibody (nAb) response rate and geometric mean (GM) ID50/ ID80 titer at enrollment by selected baseline participant characteristics.**
(DOCX)

**S2 Table. Results of multivariate modeling associating baseline participant characteristics with SARS-CoV-2 neutralizing antibody (nAb) ID80 titer at enrollment.** Each predictor of interest is studied independently for its association with the nAb ID80 titer (using log-linear regression). Each regression model adjusts for confounders: COVID-19 severity, age, sex at birth, African region, and days since SARS-CoV-2 diagnosis.
(DOCX)

**S3 Table. Estimated anti-SARS-CoV-2 neutralizing antibody (nAb) response rate and geometric mean (GM) ID50 titer by visit among people living with HIV (PLWH) and people without HIV (PWOH) by COVID-19 severity group.**
(DOCX)

**S4 Table. Estimated anti-SARS-CoV-2 neutralizing antibody (nAb) response rate and geometric mean (GM) ID80 titer by visit among people living with HIV (PLWH) and people without HIV (PWOH) by COVID-19 severity group.**
(DOCX)

## Acknowledgments

We hereby acknowledge the contribution of the HVTN 405/HPTN 1901 Study Team, which, in addition to several of the authors (Shelly Karuna, Shuying Sue Li, April K. Randhawa, Laura Polakowski, Lawrence Corey, Julia Hutter), included Shannon Grant, Stephen R. Walsh, Ian Frank, Martin Casapia, Meg Trahey, Ollivier Hyrien, Leigh Fisher, Maurine D.

Miner, James G. Kublin, David Montefiori, Katherine Gill, John Hural, William O. Hahn, Jen Hanke, Lisa Sanders, Laurie Rinn, Theresa Wagner, Doug Grove, Gail Broder, Ro Yoon, Robert De La Grecca, Carissa Karg, Alison Ayres, Vicky Kim, Megan Jones, Nick Maurice, and Simba Takuva.

We also acknowledge contributions from Lisa Bunts, Sara Thiebaud, Nicole Espy, and the HVTN 405/HPTN 1901 study participants and site investigators/research teams.

## Author contributions

**Conceptualization:** Nonhlanhla N Mkhize, Lawrence Corey, Katherine Gill, Holly Janes, Shelly Karuna.

**Data curation:** Nonhlanhla N Mkhize, Shuying Sue Li, Jiani Hu, Tandile Modise, Penny L Moore, Anton M Sholukh, David C Montefiori, Shelly Karuna.

**Formal analysis:** Nonhlanhla N Mkhize, Shuying Sue Li, Jiani Hu.

**Funding acquisition:** Lawrence Corey, Shelly Karuna.

**Investigation:** Nonhlanhla N Mkhize, Shuying Sue Li, Jiani Hu, Zaheer Hoosain, Nigel Garrett, Zvavahera M Chirenje, Llewellyn Fleurs, Haajira Kaldine, Tandile Modise, Penny L Moore, April K Randhawa, Anton M Sholukh, Julia Hutter, Laura Polakowski, Lawrence Corey, Katherine Gill, David C Montefiori, Holly Janes, Shelly Karuna.

**Methodology:** Nonhlanhla N Mkhize, Shuying Sue Li, Jiani Hu, Anton M Sholukh, Lawrence Corey, David C Montefiori, Holly Janes, Shelly Karuna.

**Project administration:** Nonhlanhla N Mkhize, Samuel T Robinson, Lawrence Corey, Shelly Karuna.

**Resources:** Nonhlanhla N Mkhize, Lawrence Corey, David C Montefiori, Shelly Karuna.

**Software:** Shuying Sue Li, Jiani Hu.

**Supervision:** Nonhlanhla N Mkhize, Shuying Sue Li, Samuel T Robinson, Zaheer Hoosain, Nigel Garrett, Zvavahera M Chirenje, Llewellyn Fleurs, Haajira Kaldine, Penny L Moore, April K Randhawa, Julia Hutter, Laura Polakowski, Lawrence Corey, Katherine Gill, David C Montefiori, Holly Janes, Shelly Karuna.

**Validation:** Shuying Sue Li, Jiani Hu.

**Visualization:** Samuel T Robinson, Holly Janes.

**Writing – original draft:** Nonhlanhla N Mkhize, Shuying Sue Li, Jiani Hu, Samuel T Robinson, April K Randhawa, Holly Janes, Shelly Karuna.

**Writing – review & editing:** Nonhlanhla N Mkhize, Shuying Sue Li, Jiani Hu, Samuel T Robinson, Zaheer Hoosain, Nigel Garrett, Zvavahera M Chirenje, Llewellyn Fleurs, Haajira Kaldine, Tandile Modise, Penny L Moore, April K Randhawa, Anton M Sholukh, Julia Hutter, Laura Polakowski, Lawrence Corey, Katherine Gill, David C Montefiori, Holly Janes, Shelly Karuna.

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
