## [Decision Letter · Decision Letter 0]

23 May 2025

PGPH-D-24-02849

Neutralizing antibody responses over time in a demographically and clinically diverse cohort of individuals recovered from SARS-CoV-2 Acquisition in Africa: A cohort study

Dear Dr. Robinson,

Thank you for submitting your manuscript to PLOS Global Public Health. After careful consideration, we feel that it has merit but does not fully meet PLOS Global Public Health’s publication criteria as it currently stands. Therefore, we invite you to submit a revised version of the manuscript that addresses the points raised during the review process.

In addition to the comment by Reviewer #1, please address the following issues:

- In the Study Cohort section, please describe how asymptomatic patients were identified and recruited.

- In the abstract, it appears that the term "univariate" is incorrectly used. If the model included multiple variables for adjustment, as indicated, it should be "multivariable" or "multivariate". Please correct or justify the use of the term "univariate".

- In Figure 1, the data points for Age Group >55 do not appear in the graph. In addition, the box for HIV = Y does not appear to be plotted. Please make these corrections.

We look forward to receiving your revised manuscript.

Kind regards,

Sanghyuk S Shin

Academic Editor

Journal Requirements:

Additional Editor Comments (if provided):

Reviewers' comments:

Reviewer's Responses to Questions

**Comments to the Author**

1. Does this manuscript meet PLOS Global Public Health’s publication criteria?

Reviewer #1: Yes

2. Has the statistical analysis been performed appropriately and rigorously?

Reviewer #1: Yes

3. Have the authors made all data underlying the findings in their manuscript fully available (please refer to the Data Availability Statement at the start of the manuscript PDF file)?

Reviewer #1: Yes

4. Is the manuscript presented in an intelligible fashion and written in standard English?

Reviewer #1: Yes

Reviewer #1: This is a thorough and well conducted study of nAb response in Africa during the initial stages of the covid-19 pandemic.

The increase in antibody response over visits 3 and 4 must be due to vaccination and infection. Is it possible to do an analysis where you look at trajectories where values prior to vaccination/infection are censored? You could also look at the effect of vaccination/infection on boosting nAbs. If you can’t do such analyses, you should list this as a limitation.

**Do you want your identity to be public for this peer review?** For information about this choice, including consent withdrawal, please see our Privacy Policy

Reviewer #1: No

---

## [Editor Report · Decision Letter 1]

5 Aug 2025

PGPH-D-24-02849R1

Neutralizing antibody responses over time in a demographically and clinically diverse cohort of individuals recovered from SARS-CoV-2 Acquisition in Africa: A cohort study

Dear Dr. Karuna,

Thank you for submitting your manuscript to PLOS Global Public Health. After careful consideration, we feel that it has merit but does not fully meet PLOS Global Public Health’s publication criteria as it currently stands. Therefore, we invite you to submit a revised version of the manuscript that addresses the points raised during the review process.

Thank you for responding to the comments from Reviewer #1 in the first review. However, it appears that you missed my comments. I have pasted them below in the additional comments section.

We look forward to receiving your revised manuscript.

Kind regards,

Sanghyuk S Shin

Academic Editor

Journal Requirements:

Additional Editor Comments (if provided):

My comments from the prior review were not addressed. Please respond to the following:

- In the Study Cohort section, please describe how asymptomatic patients were identified and recruited.

- In the abstract, it appears that the term "univariate" is incorrectly used. If the model included multiple variables for adjustment, as indicated, it should be "multivariable" or "multivariate". Please correct or justify the use of the term "univariate".

- In Figure 1, the data points for Age Group >55 do not appear in the graph. In addition, the box for HIV = Y does not appear to be plotted. Please make these corrections.
---

## [Editor Report · Decision Letter 2]

19 Aug 2025

Neutralizing antibody responses over time in a demographically and clinically diverse cohort of individuals recovered from SARS-CoV-2 Acquisition in Africa: A cohort study

PGPH-D-24-02849R2

Dear Dr. Karuna,

We are pleased to inform you that your manuscript 'Neutralizing antibody responses over time in a demographically and clinically diverse cohort of individuals recovered from SARS-CoV-2 Acquisition in Africa: A cohort study' has been provisionally accepted for publication in PLOS Global Public Health.

Best regards,

Sanghyuk S Shin

Academic Editor